

# *Lactobacillus rhamnosus* GG improves cognitive impairments in mice with sepsis

Linxiao Wang[1,*], Rui Zhao[1,*], Xuemei Li[1], Pei Shao[2], Jiangang Xie[3], Xiangni Su[2], Sijia Xu[2], Yang Huang[3] and Shanbo Hu[2,3]

[1] College of Life Sciences, Northwest University, Xi'an, Shaanxi, China
[2] School of Nursing, Air Force Medical University, Xi'an, Shaanxi, China
[3] Department of Emergency, Xijing Hospital, Air Force Medical University, Xi'an, Shaanxi, China
[*] These authors contributed equally to this work.

Corresponding authors
Yang Huang, huangy@fmmu.edu.cn
Shanbo Hu, hushanbo122@126.com

## ABSTRACT

**Background**. Survivors of sepsis may encounter cognitive impairment following their recovery from critical condition. At present, there is no standardized treatment for addressing sepsis-associated encephalopathy. *Lactobacillus rhamnosus* GG (LGG) is a prevalent bacterium found in the gut microbiota and is an active component of probiotic supplements. LGG has demonstrated to be associated with cognitive improvement. This study explored whether LGG administration prior to and following induced sepsis could ameliorate cognitive deficits, and explored potential mechanisms.
**Methods**. Female C57BL/6 mice were randomly divided into three groups: sham surgery, cecal ligation and puncture (CLP), and CLP+LGG. Cognitive behavior was assessed longitudinally at 7-9d, 14-16d, and 21-23d after surgery using an open field test and novel object recognition test. The impact of LGG treatment on pathological changes, the expression level of brain-derived neurotrophic factor (BDNF), and the phosphorylation level of the TrkB receptor (p-TrkB) in the hippocampus of mice at two weeks post-CLP (16d) were evaluated using histological, immunofluorescence, immunohistochemistry, and western blot analyses.
**Results**. The CLP surgery induced and sustained cognitive impairment in mice with sepsis for a minimum of three weeks following the surgery. Compared to mice subjected to CLP alone, the administration of LGG improved the survival of mice with sepsis and notably enhanced their cognitive functioning. Moreover, LGG supplementation significantly alleviated the decrease in hippocampal BDNF expression and p-TrkB phosphorylation levels caused by sepsis, preserving neuronal survival and mitigating the pathological changes within the hippocampus of mice with sepsis. LGG supplementation mitigates sepsis-related cognitive impairment in mice and preserves BDNF expression and p-TrkB levels in the hippocampus.

## INTRODUCTION

Sepsis is a life-threatening pathological condition caused by infection-induced dysregulated host response (*Singer et al., 2016*). Sepsis is highly prevalent with a high mortality rate,

*e.g.*, in 2017 it affected 48.9 million people worldwide and represented almost 20% of global deaths (*Rudd et al., 2020*). Sepsis remains one of the major contributors to mortality in severely ill patients (*Chung et al., 2020*). Sepsis survivors can experience short or long term neurocognitive dysfunction, also referred to as sepsis-associated encephalopathy (SAE), which can deteriorate and become life-threatening (*Tauber et al., 2021*). Currently, there is no standardized treatment for cognitive impairment in clinical sepsis survivors.

The pathogenesis of SAE is not clearly understood. Several factors contribute to the development of SAE, including impaired end-organ perfusion-related hypoxia, oxidative stress, dysregulated inflammation, cell injury in the brain, damage in the blood–brain barrier (BBB), mitochondrial dysfunction, toxic neuropeptide accumulation, and impaired toxin clearance (*Barlow et al., 2022*). Furthermore, the use of broad-spectrum antibiotics causes changes in the gut microbiome, possibly contributing to the development of SAE in patients with sepsis (*Adelman et al., 2020*). Treatment with fecal microbiota transplantation (FMT) has been shown to improve survival and reduce symptoms of encephalopathy in patients with sepsis (*Fang et al., 2022*). Similarly, treatment with short-chain fatty acids from gut microbiota can improve the prognosis of SAE (*Zhang et al., 2023*), and treatment with 2% hydrogen (H2) inhalation or hydrogen-rich water can attenuate SAE-related cognitive impairment, improving gut flora dysbiosis and partially correcting metabolic disturbances in patients with sepsis (*Han et al., 2023*).

*Lactobacillus rhamnosus* GG (LGG) is a common bacterium in the gut microbiota and in probiotic supplements. LGG has been used as in intervention in several diseases in the gastrointestinal system, in metabolic disorders, and in dysregulated immune responses (*Capurso, 2019*). LGG can reduce intestinal bacterial translocation and improve intestinal bacterial homeostasis in rats, modulating the gut-brain axis and ameliorating cognitive deficits (*Li et al., 2023*). However, it is still unclear whether LGG treatment can improve SAE-related cognitive deficits. Additionally, the current research on sepsis and SAE mainly focuses on male mice. Few studies on sepsis and SAE have been performed in female mice, which is an important gap in the current research as gender differences may affect learning and memory behavior (*Fleischer & Frick, 2023*). This study explored whether continuous daily administration of LGG prior to and following induced sepsis could ameliorate cognitive deficits in female mice. This study also explored potential mechanisms of LGG in sepsis-associated encephalopathy.

## MATERIALS & METHODS

### Mice

Female C57BL/6 mice (8-week-old, 18–22 g) were obtained from the Animal Center of the Air Force Medical University (Xi'an, China), which were randomly divided into control (24 mice) and surgical (120 mice) groups. These animals were raised in an animal facility with five to six animals per cage, a constant temperature of 22 °C, relative humidity of 45%–55%, and a 12-h light/dark cycle. Mice were allowed to unlimited access to normal chow and water *ad libitum*, and they were acclimated for at least 10 days before surgery was performed. At the end of the experiment, the mice were euthanized with carbon dioxide:

the cages were filled with CO2 at an equilibrium rate based on a CO2 (purity greater than 99%) replacement rate of 30% to 70% per minute of the container volume, to prevent sickness before euthanasia. All animal experiments were performed in accordance with the laboratory animal care and guidelines of the Air Force Medical University, and the experimental protocol was approved by the Welfare and Ethics Committee of the Air Force Medical University, Xi'an, Shaanxi, China (Approval date: 30/06/2023; Approval No. IACUC-20230038).

## CLP-induced sepsis

Cecal ligation and puncture (CLP) surgery is the gold standard for mouse sepsis models and is effective in inducing short- and long-term behavioral disturbances (*Savi et al., 2021*). Surgery was performed on the mice in this study to establish CLP-induced sepsis, as described by *Rittirsch et al. (2009)*. Mice were anesthetized using 3% isoflurane gas, and a small incision (1 cm in length) was cut in their left lower abdomen near the cecum. The cecum was then gently removed and ligated with a 6-0 thread about 50% below the ileocecal valve, and the end of the cecum was perforated with a 21G needle, extruding a small amount of feces to induce moderate degrees of sepsis (*Zhu et al., 2023*). Mice in the control group received sham surgery without cecum ligation and perforations. All mice were postoperatively injected with 50 mL/kg saline subcutaneously on the back to compensate for fluid loss during the surgery. The general behavioral and survival rate of mice were also assessed postoperatively.

## Treatments

*Lactobacillus rhamnosus* GG (LGG, ATCC 53103) was purchased from Bnbio in Beijing, China and anaerobically cultured in Lactobacilli MRS Broth (288130, BD Difco, Franklin Lakes, NJ, USA; *Chen et al., 2023*). After washing the fermentation solution with sterile saline three times, the concentration of the solution was adjusted to $5 \times 10^9$ CFU/mL using sterile saline. Mice were divided into three groups: sham surgery, CLP, and CLP+LGG. Mice were administered either 200 µL of a bacterial solution (LGG) or an equivalent amount of normal saline (vehicle) using a No. 12 gavage needle at the same time every day, beginning one week prior to surgery (-7d) and continuing for one, two, or three weeks postoperatively (7/14/21d; Fig. 1A). At the end of the treatment, the mice in all three groups (sham, CLP, and CLP+LGG) were performed behavioral testing at three time points: one week, two weeks, and three weeks after surgery, and samples were taken immediately after the behavioral tests were completed.

## Open field test

The open field test (OFT) is an assay used to evaluate the autonomous and exploratory behavior of mice in novel or unfamiliar environments. If the mice were repeatedly exposed to the same open field, their activity levels were generally lower compared to the first exposure to the open field due to their increased familiarity with the open field area. Therefore, each mouse was subjected to behavioral testing at only one time point. Before the OFT, all mice were acclimatized to the surrounding environment for 1 h. The OFT was performed in an open-air, square box (50 cm × 50 cm × 40 cm) with uniform light

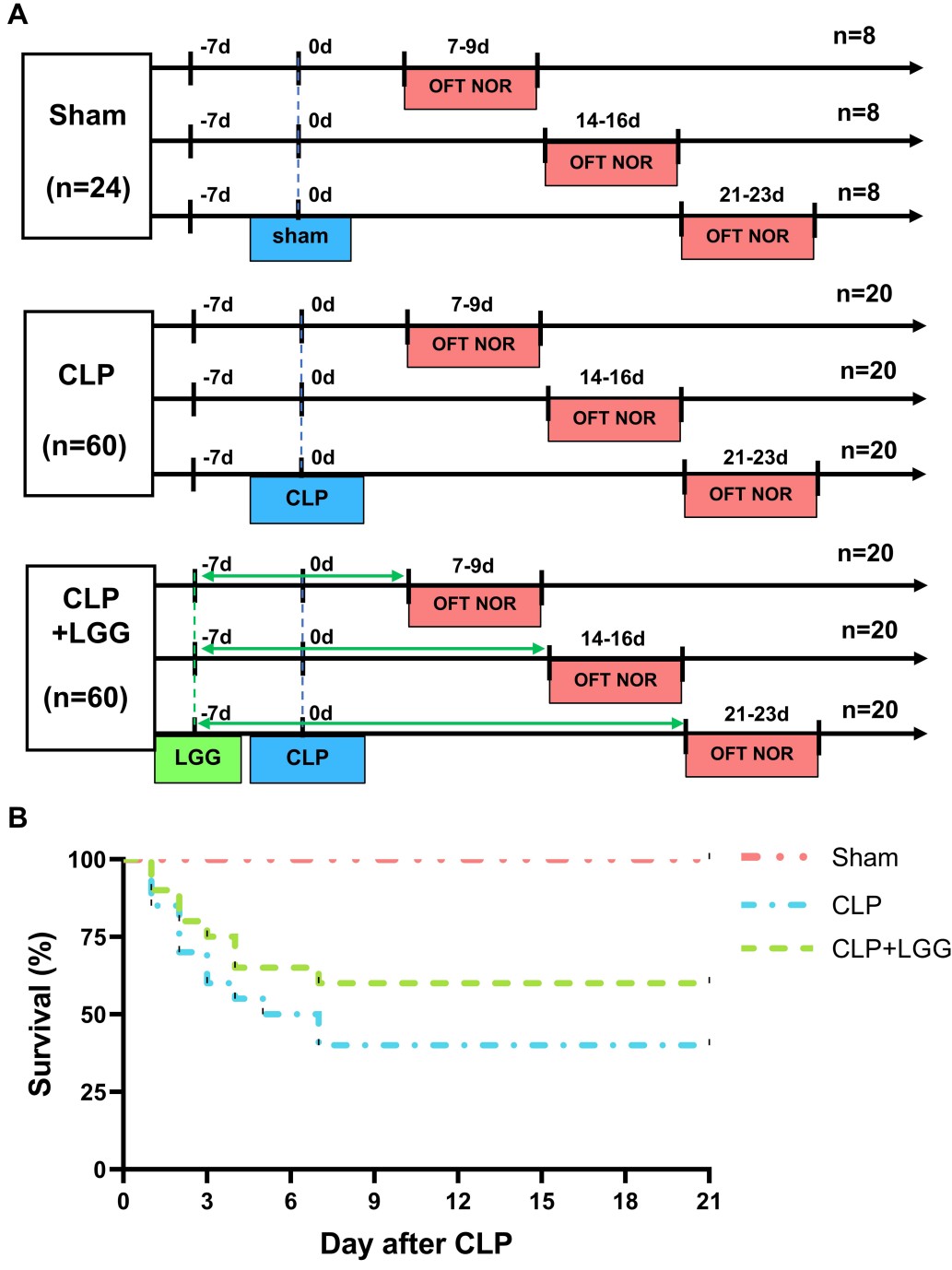

**Figure 1 Behavioral test lines and survival curves of different groups of mice.** (A) Behavioral testing line and treatment schedule for probiotic administration. (B) Survival curves of mice in different treatment groups (n = 20/group).

in each corner. The open box was divided into 16 equal compartments, and the middle four compartments were defined as the central area. During the experiment, each mouse was placed in the center of the open box with its back to the experimenter and allowed to freely explore for 10 min. The mouse was monitored by videorecording and the traveling distance and the time in the central area were analyzed using tracking software (Noldus EthoVision).

## Novel object recognition test

The Novel object recognition test (NORT) is a learning memory test that simulates the learning memory behavior of animals by taking advantage of their innate tendency to explore new objects. The NORT is divided into adaptation, training, and test phases. On the first day, mice were allowed to freely explore in an open box for 10 min (adaptation phase). At the same time on the second day, each mouse was exposed to two identical objects for 10 min in the center of the box (training phase). After an hour of inter-trial intervals (ITI), the mouse was tested to see how it explored the familiar object (A1) and a completely different object (A2) in the box, and the mouse's movement trajectory was recorded for 10 min. The objects and the field were washed with 75% ethanol before each trial and between each mouse. Exploration time with each time was recorded when the mouse touched the object or its head moved towards the object (within two cm). Preference index refers to the time spent exploring the new object (A2) divided by the total time spent exploring both objects (A1+A2).

## Western blotting

Hippocampal tissues were dissected from individual mice and homogenized in lysis buffer. After centrifugation, the concentrations of total proteins in tissue lysates were quantified using the BCA kit. The tissue lysates (50 µg/lane) were separated by sodium dodecyl-sulfate polyacrylamide gel electrophoresis (SDS-PAGE) and were subsequently transferred onto polyvinylidene difluoride (PVDF) membranes. The membranes were blocked with 5% BSA in TBST and probed with anti-BDNF (#ab108319, Abcam, Cambridge, UK), anti-TrkB (#4603S, Cell Signaling Technology, Danvers, MA, USA), anti-p-TrkB (#bsm-52213R, Bioss Antibodies, Beijing, China), or anti- $\beta$-Actin (#4967, Cell Signaling Technology, Danvers, MA, USA) overnight at 4 °C. After being washed, the membranes were reacted with appropriate HRP-labeled secondary antibodies, and were then developed with chemiluminescent reagents and Western Lumax Light (Zeta Life, New York, NY, USA). The data were analyzed using the Chemi Doc MP imaging system (Bio-Rad, Hercules, CA, USA) and ImageJ software.

## Pathological staining

Mice were anesthetized and cardiac-perfused with 4% paraformaldehyde and phosphate-buffered saline (PBS). Their brain tissues were dissected and fixed in 4% paraformaldehyde for 72 h and paraffin-embedded. The brain tissue sections (5 µm) were routine-stained with Nissl, hematoxylin-eosin (HE), and subjected to immunofluorescence (IF) and immunohistochemistry (IHC). For IF, the sections were incubated with anti-BDNF or anti-NeuN antibody (#GB11138, Servicebio, Wuhan, China) overnight at 4 °C, and

reacted with fluorescent dye-coupled secondary antibodies, followed by nuclear staining with 4′,6-diamidino-2-phenylindole (DAPI). The stained sections were examined and observed under a fluorescent microscope. For IHC, the paraffin-embedded brain tissue sections were dewaxed, rehydrated, treated with 3% hydrogen peroxide in methanol, and subjected to antigen retrieval by heating them in citric acid solution. The sections were incubated overnight in a cool room with primary anti-BDNF antibody and reacted with HRP-labeled secondary antibodies. Visualization was performed using the DAB color development kit (#G1212, Servicebio, Wuhan, China), and immunosignals were observed under a light microscope.

## Statistical analysis

This study was performed by researchers following the principles of randomization and blinding: the mice were randomly divided into either the control group or the experimental groups through a random number table, and the researchers engaged in the experiment and data analysis did not know which mice were experimental mice and which were control mice. Neuronal cell counts, immunofluorescence intensities, and grayscale analysis of western blot bands were analyzed with Image J software. Data were expressed as mean ± standard error of the mean (SEM) and were statistically analyzed by one-way and two-way analysis of variance (ANOVA) using GraphPad Prism 8.0 software (GraphPad Software Inc., San Diego, CA, USA). Statistically significant differences were defined as $P$-value $< 0.05$.

# RESULTS

## LGG improves survival rate in CLP mice

The 21-day survival rate of mice with moderate sepsis (CLP group) was 40%, whereas the survival rate of LGG-treated mice with sepsis (CLP+LGG group) was 60% at the end of the experiment (Fig. 1B), indicating that LGG improved the survival of mice with sepsis.

## LGG ameliorates cognitive impairment and emotional abnormalities in CLP mice

The effects of CLP-associated sepsis on the cognitive and emotional behaviors of individual mice were examined longitudinally by OFT and NORT after CLP surgery. Autonomous and emotional behaviors were analyzed by examining the behavioral trajectories of the mice and their dwell times within the open field region (OFT; Fig. 2A). Quantitative analysis showed that there was no significant difference in the total distance traveled between the three groups of mice ($P > 0.05$, Fig. 2B), which indicated that CLP surgery did not significantly affect the locomotor status of the mice. However, from 7d to 21d after surgery, mice in the CLP group traveled significantly less distance and spent significantly less time in the central region than those in the sham operation group ($P < 0.01$, Figs. 2C, 2D). LGG-treated CLP mice did not show significant differences in locomotion within the open field area from the CLP group mice until 7d after surgery; between 14d and 21d after surgery, CLP+LGG mice traveled significantly farther and spent significantly more time in the central region than the CLP group mice ($P < 0.05$, Figs. 2C, 2D). There was decreased

time and distance in the inner area of the OFT, in the CLP mice, which indicates the CLP-induced sepsis caused increased anxiety-like behavior, and this was ameliorated in the mice with LGG supplementation. The NORT uses the innate preference for novelty in mice to reveal a prior encounter with an object in memory, reflecting the function and relative health of specific brain regions involved in memory in mice (Fig. 3A). The NORT results showed that mice in the CLP group showed a decreased interest in novel objects, with a significant decrease in the time taken to touch the novel object and the preference index from 9d to 23d after surgery ($P < 0.05$, Fig. 3B). These data suggest that CLP-associated sepsis causes cognitive impairment in mice. LGG supplementation mice with sepsis also showed significantly increased time to touch new objects in the NORT 16d after surgery compared with the CLP group ($P < 0.01$, Fig. 3B); moreover, by 23d after surgery, mice in the LGG group had comparable times to touch new objects and comparable cognitive indices to those of the sham group, both of which were significantly larger than those of the CLP group ($P < 0.05$, Fig. 3B). In conclusion, LGG supplementation improved cognitive impairment in mice with sepsis.

## LGG ameliorates pathological injury in CLP mice

The formation of cognitive functions is a complex process involving various regions of the brain, including the hippocampus. The hippocampus plays a crucial role in learning and memory, synaptic plasticity, and neurodevelopment. In order to understand the therapeutic effect of LGG, pathologic examination of mouse brain tissue was performed 16d after surgery, and HE staining and Nissl staining were used to observe the morphological changes of the cornu ammonis 1 (CA1), cornu ammonis 3 (CA3), and dentate gyrus (DG) regions in the different groups of mice (*Kajikawa et al., 2022*). HE results showed that in the CA1 region and CA3 region of the sham operation control group, the nuclei were clear and the neuronal cells were neatly arranged and tight. The neurons in the CA1 and CA3 areas of the hippocampus of CLP-induced mice with sepsis were disorganized, the nuclei of the neurons were consolidated, some neurons were lost, and the space around the cells was enlarged. In CLP-induced mice, many granule cells in the dentate gyrus (DG) area had ill-defined nuclei and there were significantly more necrotic cells than in the sham group. In the mice with sepsis that were treated with LGG, hippocampal neurons were arranged normally, with a small number of necrotic neurons, and neuronal cell damage was significantly reduced (Fig. 4A). Nissl staining showed a significantly reduced number of pyramidal cells in the CA1 and CA3 regions of CLP mice compared with sham group; however, this damage was partially mitigated in LGG-treated mice, and the Nissl vesicles were uniformly distributed, which significantly prevented Nissl vesicle and neuronal loss (Fig. 4B). There was no significant difference in the number of live neurons in the hippocampal dentate gyrus (DG) region of mice in each group (Figs. 4C–4E). These results all indicate LGG supplementation preserved neuronal survival in mice with sepsis.

## LGG administration enhances BDNF expression and p-TrkB to preserve neuronal survival in CLP mice

BDNF is a neurotrophic factor that regulates neuronal regeneration and synaptic tangibility, promoting hippocampal neuronal regeneration and improving learning and memory

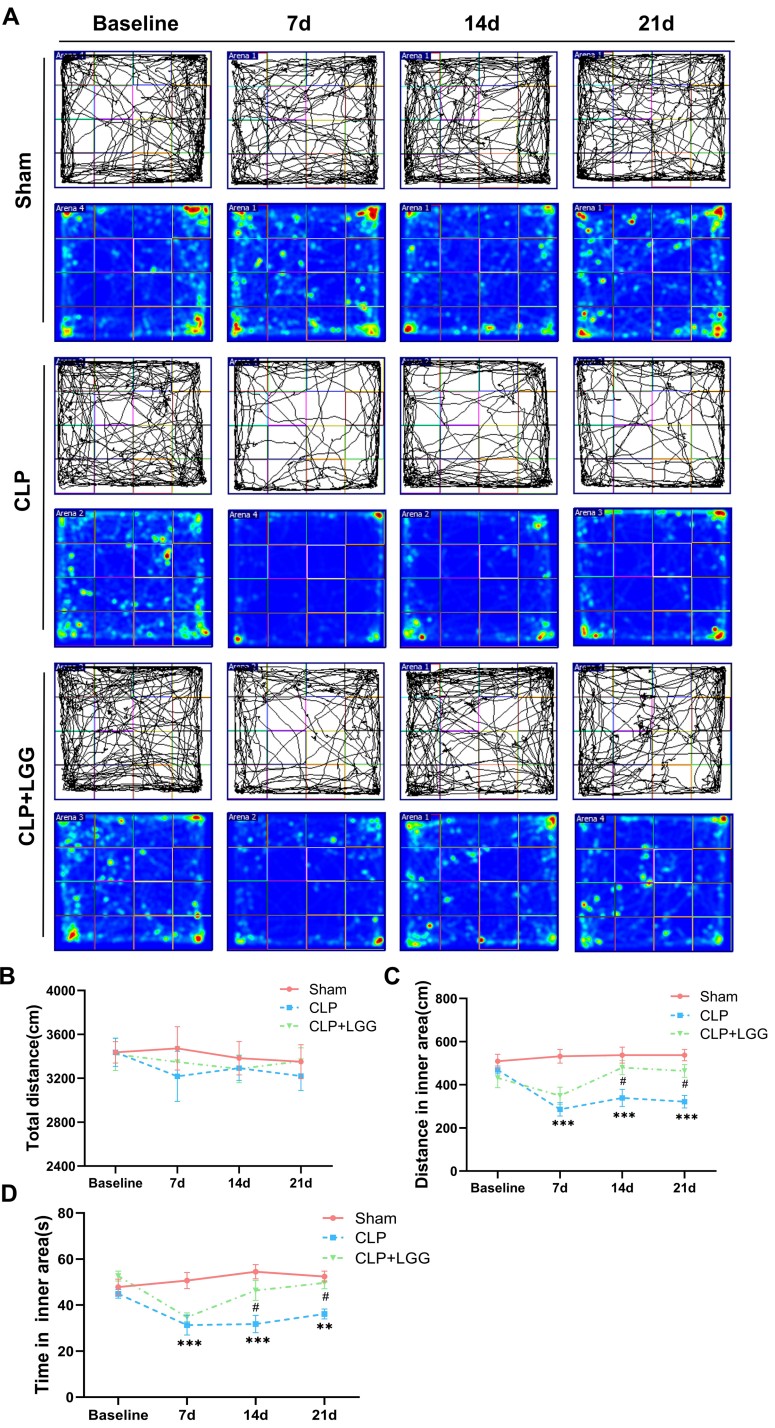

**Figure 2** **LGG supplementation alleviates emotional abnormalities in mice with sepsis.** (A) Behavioral trajectories and thermograms of mice tested in the open field 0, 7, 14, and 21 days after surgery. (B) Total distance moved by mice in the open field. (C, D) Distances and times of mice moving in the central area of the open field. Data are expressed as mean ± SEM (n = 8/group). **$P < 0.01$ *vs.* sham surgery group; *** $P < 0.001$ *vs.* sham surgery group; # $P < 0.05$ *vs.* CLP group.

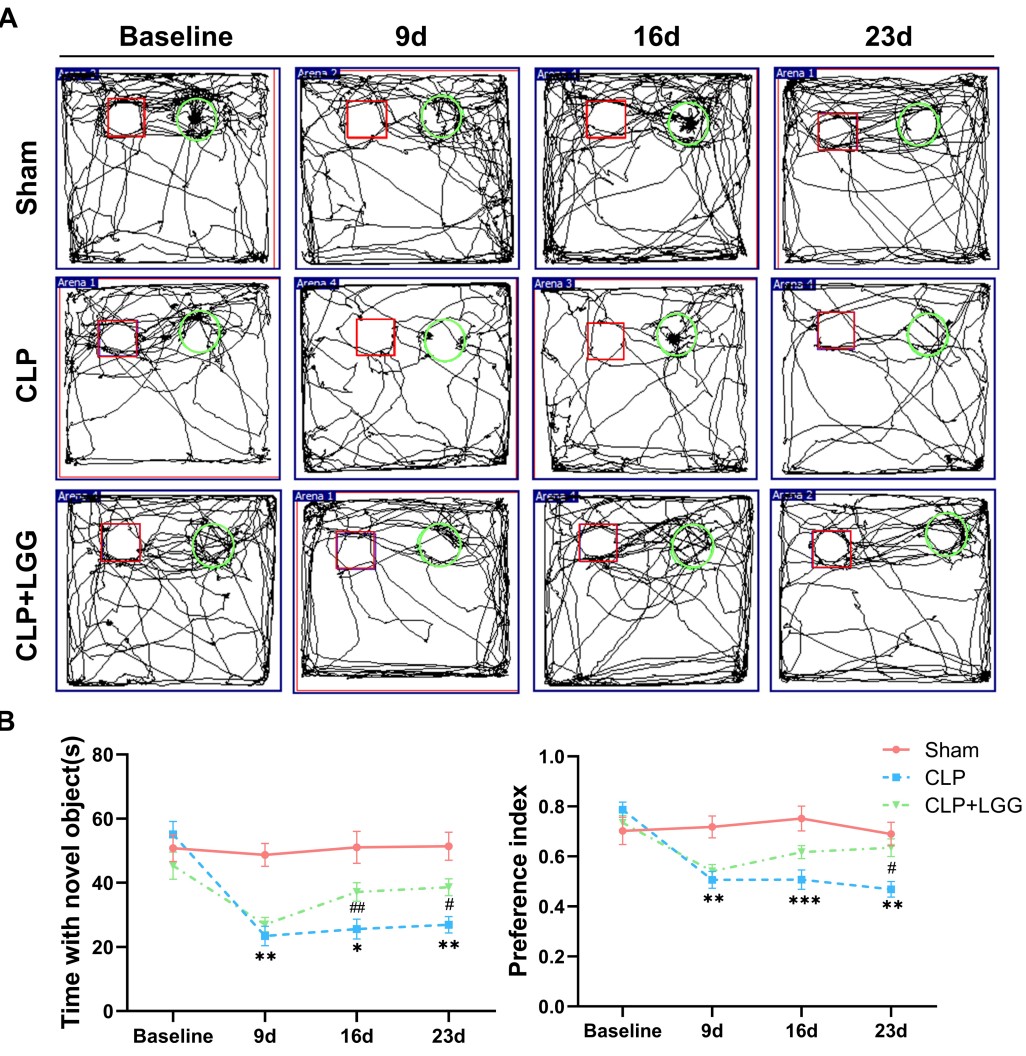

**Figure 3** **LGG supplementation improves cognitive dysfunction in mice with sepsis.** (A) Trajectory diagram of mouse movements in the new object recognition test, with boxes indicating old objects and circles indicating new objects. (B) Exploration time and preference index of the new object in the new object recognition test. Data are expressed as mean ± SEM (n = 8/group). *$P < 0.05$ *vs.* sham surgery group; **$P < 0.01$ *vs.* sham surgery group; ***$P < 0.001$ *vs.* sham surgery group; # $P < 0.05$ *vs.* CLP group; ## $P < 0.01$ *vs.* CLP group.

functions. Neuron-specific nuclear protein (NeuN) is a neuron-specific RNA-splicing protein involved in neuronal developmental differentiation and synaptogenesis. The expression level of NeuN has been used to assess neuronal death or loss. This study examined the expression of BDNF in neuronal cells using immunofluorescence and immunohistochemistry. IF showed that the expression of BDNF in neurons in the hippocampal CA1 and DG regions was significantly lower in the CLP group compared with the sham operation group (Figs. 5A, 5B), and the number of NeuN-labeled mature neurons was significantly reduced (Figs. 5C, 5D). In contrast, LGG-treated mice with sepsis partially recovered the reduced expression of BDNF and NeuN (Figs. 5C, 5D).

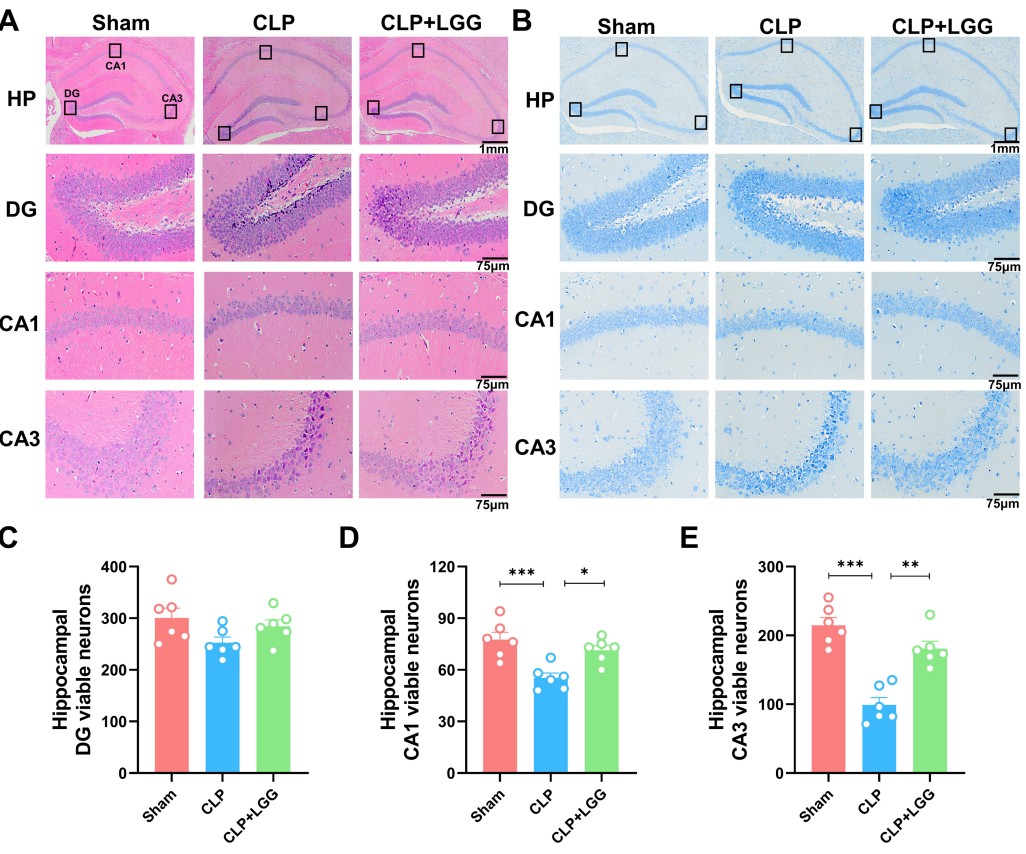

**Figure 4  Pathological changes in the hippocampal region of mice with sepsis supplemented with LGG.**
(A) HE staining of the HP, DG, CA1, and CA3 regions in the three groups of mice. (B) Nissl staining of the HP, DG, CA1, and CA3 regions in the three groups of mice. HP scale: one mm ; DG, CA1, CA3 scale: 75 μm. (C) The number of neurons visible in the DG area. (D) The number of neurons visible in the CA1 area. (E) The number of visible neurons in the CA3 area. Data are expressed as mean ± SEM (n = 6/group). *$P < 0.05$, **$P < 0.01$, ***$P < 0.001$. HP, hippocampus; CA1, cornu ammonis 1; CA3, cornu ammonis 3; DG, dentate gyrus.

The overall reduction of BDNF expression in the cortex and hippocampus of CLP mice was detected by IHC (Figs. 6A, 6B). Western blotting showed that BDNF expression and TrkB phosphorylation (p-TrkB) levels in the hippocampal region of CLP mice were significantly reduced compared with those of the sham group, but BDNF expression and TrkB phosphorylation (p-TrkB) levels were almost restored in the mice of the CLP+LGG group (Figs. 6C, 6D). Thus, LGG attenuates SAE-reduced BDNF expression and p-TrkB levels to maintain neuronal survival and ameliorate cognitive impairment in mice with sepsis.

## DISCUSSION

A previous study demonstrated that probiotic supplementation can improve cognitive dysfunction in children and adolescents and reduce the risk of attention deficit hyperactivity disorder (ADHD) and Asperger syndrome (AS; *Rianda et al., 2019*). LGG is an abundant

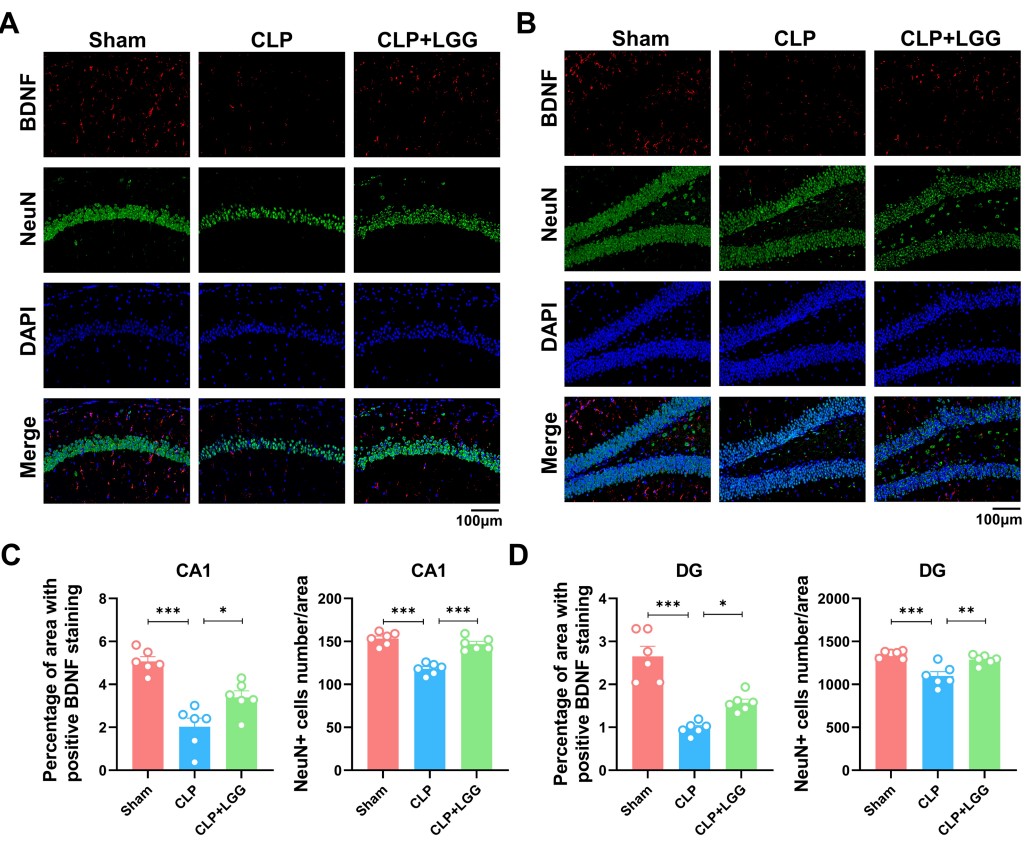

**Figure 5** **Effects of LGG supplementation on BDNF and NeuN expression level in the hippocampal region of CLP mice.** (A) Immunofluorescence of BDNF and NeuN in the hippocampal CA1 region. (B) Immunofluorescence of BDNF and NeuN in the DG region of hippocampus. Scale bar = 100 μm. (C) Quantitative analysis of the number of neuronal cells and the proportion of BDNF staining in the CA1 region of the hippocampus. (D) Quantitative analysis of the number of neuronal cells and the proportion of BDNF staining in the DG region of the hippocampus. Data are expressed as mean ± SEM (n = 6/group). *$P < 0.05$, **$P < 0.01$, ***$P < 0.001$. BDNF, brain-derived neurotrophic factor; NeuN, neuron-specific nuclear protein.

strain of bacterium in the gut microbiota and in probiotic supplements, and it is relatively safe for human application (*Capurso, 2019*). Studies have shown that LGG can benefit children with diarrhea caused by long-term treatment with antibiotics (*Goldenberg et al., 2015*) as well as very low birth weight newborns with necrotizing small intestinal colitis (*Manzoni et al., 2014*). LGG has also been used to inhibit sepsis-related inflammation to protect the septic intestinal mucosal barrier and improve organ damage (*Chen et al., 2023*). The present study also found that the 21d survival rate of mice with sepsis was about 40%, while LGG supplementation improved the 21d survival rate of mice with sepsis to 60%, which is consistent with previous findings (*Tsui et al., 2021*) and may be related to the homeostasis of the microbiome-gut-brain axis (*Giridharan et al., 2022*). Sepsis can lead to blood–brain barrier breakdown, neuroinflammation, neurotransmitter dysfunction, and neuronal loss (*Li, Ji & Yang, 2022*; *Annane & Sharshar, 2015*). Psychiatric behavioral abnormalities can severely affect the quality of life and prognosis of patients with sepsis,

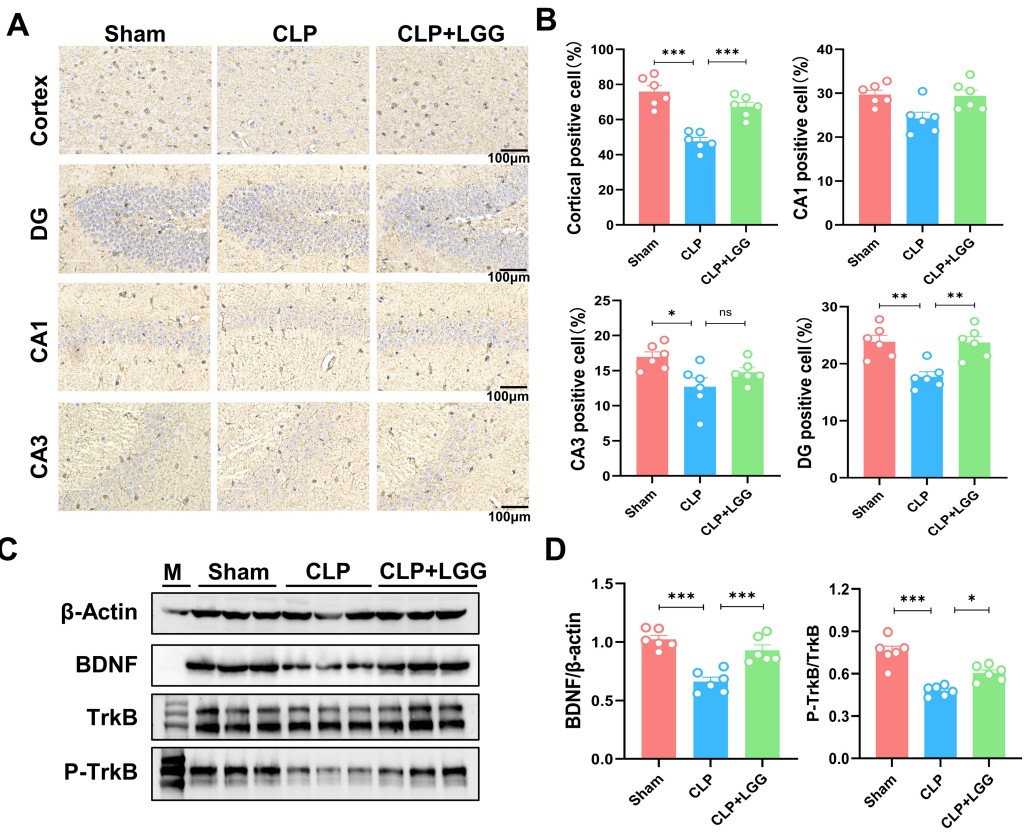

**Figure 6** **LGG promotes BDNF expression level in the hippocampal region of CLP mice.** (A) Immuno-histochemistry of BDNF in cortical and hippocampal regions. Scale bar = 100 μm. (B) Statistical analysis of immunohistochemistry of BDNF. (C) Immunoblot analysis of protein expression of BDNF, TrkB , and p-TrkB. (D) Quantification of BDNF protein expression and p-TrkB/TrkB protein content ratio. Data are expressed as mean ± SEM (n = 6/group). *$P < 0.05$, **$P < 0.01$, ***$P < 0.001$, ns = no statistical significance.

and there is currently no effective treatment for SAE. This study demonstrated that LGG administration improved cognitive dysfunction in mice in the early stages of sepsis.

The severity and duration of sepsis impact the risk of sepsis-associated mental or behavioral abnormalities, which can manifest as anxiety, depression, and impairment in cognition, learning, and memory. When these psychobehavioral abnormalities occur in patients with sepsis, the symptoms and complications associated with the disease worsen. Anxiety and cognitive deficits are the main manifestations of sepsis-related cognitive deficits (*Helbing, Böhm & Witte, 2018*). This study first investigated the neuroprotective effects of LGG on cognitive deficits in sepsis-surviving mice using relevant behavioral tests such as the OFT and the NOR. The results showed that sepsis-surviving mice developed abnormal behavioral responses, such as decreased time in the central square (OFT) and low interest in novel objects (NOR). These abnormal responses manifested 7d after surgery and persisted 23d after the procedure, suggesting that sepsis induced significant cognitive impairment, which is consistent with the findings of previous research (*Tian et al., 2023*).

LGG administration significantly ameliorated cognitive deficits in CLP-induced 2-week sepsis-surviving mice, as evidenced by increased time spent in the center square and increased interest in new objects. In addition, studies were conducted to observe neuronal damage in the hippocampus. Disturbed neuronal arrangement and increased necrotic cells were found in the hippocampus on day 16 after CLP-induced sepsis. LGG significantly prevented loss of Nissl vesicle and neuronal loss and significantly improved cognitive impairment in mice with sepsis. These findings may help ameliorate psychobehavioral abnormalities in patients with sepsis. LGG is the most common probiotic and has been widely used in clinical settings (*Szajewska & Hojsak, 2020*). Recent studies have found that promoting LGG intestinal colonization could improve noise-induced cognitive deficits and systemic inflammation in rats (*Li et al., 2023*). These results support the notion that LGG may effectively improve psychobehavioral abnormalities by modulating the gut-brain axis.

This study revealed that LGG administration significantly ameliorated sepsis-induced pathological changes and improved neuron survival in the hippocampus of mice. Hippocampal neuron and BDNF expression were measured to assess neuronal injury and to determine whether supplementation with LGG ameliorates neuroinflammation and sepsis. The results showed that LGG supplementation significantly mitigated the sepsis-decreased BDNF expression level and p-TrkB in the hippocampus of mice with sepsis, which is consistent with previous reports (*Cheng et al., 2020*; *Liu et al., 2022*; *Orlando et al., 2020*). BDNF is a neuronal growth and survival factor and is crucial for neuronal synaptic plasticity (*Kowianski et al., 2018*; *Wang, Kavalali & Monteggia, 2022*). LGG's preservation of BDNF expression level may contribute to the protection against neuronal death observed in the hippocampus of LGG-treated mice with sepsis. The expression levels of BDNF and its receptor and signaling are also important in the development and progression of Alzheimer's disease (AD). BDNF is highly expressed in the hippocampus and cortex. BDNF also binds to its high affinity TrkB receptors and regulates synaptic function and neuronal survival (*Zhao et al., 2022*).

BDNF is crucial to the synaptic remodeling of cognitive processes and may serve as a biomarker for cognitive function. BDNF expression level is correlated with the degree of cognitive dysfunction (*Siuda et al., 2017*). Alterations in the BDNF/TrkB signaling pathway with aging, and pathological conditions are all potential mechanisms of cognitive impairment (*Numakawa & Odaka, 2022*). Therefore, decreased BDNF expression levels in the hippocampus may contribute to the pathogenesis of cognitive dysfunction, and supplementation with LGG to preserve BDNF expression level and its related signaling through the gut-brain axis may promote neuronal cell plasticity and ameliorate cognitive-behavioral deficits in mice with sepsis.

## CONCLUSION

The early cognitive impairment seen in mice with sepsis is related to the sepsis-induced downregulation of BDNF expression in the hippocampus. LGG supplementation mitigates sepsis-related cognitive impairment and preserves BDNF expression and p-TrkB levels in the hippocampus of mice with sepsis. These preliminary findings from a surgically-induced

sepsis mouse model may facilitate the design of clinical studies on the effects of LGG on cognitive dysfunction in patients with sepsis.

### Funding

This work was supported by the Key Research and Development Program of Shaanxi Province under Grant (number 2022JQ-852); the Natural Science Foundation Project in Shaanxi Province under Grant (number 2023-JC-YB-686); and the General Project Surface Project in Shaanxi Province under Grant (number 2022JM-566). The Medjaden Academy & Research Foundation for Young Scientists (Grant No. MJA 202306018) provided funding for the language polishing of the manuscript. The funders had no role in study design, data collection and analysis, decision to publish, or preparation of the manuscript.

### Grant Disclosures

The following grant information was disclosed by the authors:
The Key Research and Development Program of Shaanxi Province:  2022JQ-852.
The Natural Science Foundation Project in Shaanxi Province: 2023-JC-YB-686.
The General Project Surface Project in Shaanxi Province: 2022JM-566.
The Medjaden Academy & Research Foundation for Young Scientists: MJA 202306018.

### Competing Interests

The authors declare there are no competing interests.

### Author Contributions

- Linxiao Wang conceived and designed the experiments, performed the experiments, analyzed the data, prepared figures and/or tables, authored or reviewed drafts of the article, and approved the final draft.
- Rui Zhao conceived and designed the experiments, performed the experiments, analyzed the data, prepared figures and/or tables, authored or reviewed drafts of the article, and approved the final draft.
- Xuemei Li conceived and designed the experiments, performed the experiments, analyzed the data, prepared figures and/or tables, and approved the final draft.
- Pei Shao performed the experiments, authored or reviewed drafts of the article, and approved the final draft.
- Jiangang Xie conceived and designed the experiments, analyzed the data, prepared figures and/or tables, and approved the final draft.
- Xiangni Su performed the experiments, authored or reviewed drafts of the article, and approved the final draft.
- Sijia Xu performed the experiments, authored or reviewed drafts of the article, and approved the final draft.
- Yang Huang analyzed the data, authored or reviewed drafts of the article, and approved the final draft.

- Shanbo Hu conceived and designed the experiments, prepared figures and/or tables, authored or reviewed drafts of the article, and approved the final draft.

## Animal Ethics

The following information was supplied relating to ethical approvals (i.e., approving body and any reference numbers):

The Welfare and Ethics Committee of the Air Force Medical University.

## Data Availability

The raw measurements are available in the Supplementary Files.

## Supplemental Information

Supplemental information for this article can be found online at http://dx.doi.org/10.7717/peerj.17427#supplemental-information.

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
