# Peer review of "Lactobacillus rhamnosus GG improves cognitive impairments in mice with sepsis"

_PeerJ, doi:10.7717/peerj.17427_

## Round 0.1 · original submission · Major Revisions

General:

Overall this appears to be a well designed and carefully conducted study likely to be of interest. However, both reviewers have noted issues that will need to be addressed before the manuscript could be accepted for publication.

I have the following additional comments for the authors to consider:

Throughout the manuscript the authors refer to "septic mice" and/or "septic patients". The terms "mice with sepsis" or "patients with sepsis" are more accurate and appropriate.

While generally well written, the article does have many errors associated with the use of English. For example:
Line 51: "escape from" would be better phrased as "avoid"
Line 56: "The pathogenesis of SAE has not been clarified.." would be better phrased as: "The pathogenesis of SAE is not clearly understood.."
Line 61: "changes in the gut microbiome" should either be "leads to changes in the gut microbiome" or "changes the gut microbiome".
Line 229: "LGG has exhibited to benefit children.." should be reworded, for examples: "Studies have shown that LGG has benefits for children.."

There are many other similar errors. I recommend that the authors have their manuscript proof-read by a native English speaker, ideally using a professional English language editing service.

Abstract:

Line 23: It is stated that "At present, there is no treatment for addressing sepsis-associated encephalopathy." However, in the introduction (line 54) it says "Currently, there is no standardized treatment for cognitive impairment in clinical sepsis survivors." I suggest the authors also use the term "standardized" in the abstract for consistency.

Materials & Methods:

Line 80: Why were female mice used? It would be helpful to have a justification for this decision.

Line 100: As Reviewer 1 has noted, the experimental design and animal numbers are not particularly clear. It would seem that there were two sets of experiments: The first involved testing the impact of the CLP procedure at three different time points (results shown in Figure 1) and the second tested the LGG supplementation at 2 weeks. This should be made clearer in the methods, as should the total number of mice used for each of the treatments across all of the experiments.

Line 147: The authors should provide information on how they assessed any morphological changes. For example, were cells counted within the defined regions of the prepared slides? Was this done by researchers blinded to treatment?

Results:

Reviewer 2 has raised concerns regarding the validity of the data. While I do not agree with all of these (e.g., the comments regarding BNDF, as noted in my following point), the authors must address these concerns. This could include (for example) clearly highlighting key differences between the different treatments within the images shown in Figure 3 (perhaps examples of well-defined nuclei, and altered size and cellular arrangement of Nissl vesicles). Better describing their methods for assessing these outcomes (as I have noted above) would also help (see also the comment from Reviewer 2 re: counting viable neurons).

BNDF: Reviewer 2 states that they do not see any difference CLP and CLP+LGG for BDNF expression. While I do not agree with this comment (particularly for Figure 4B, which seems to show a clear difference between those two groups) I think that, as already noted, a clearer description of how these differences were assessed and quantified would be helpful in addressing the concerns raised by this reviewer.

Discussion:

Line 246: "may assist scholars in preventing psychobehavioral abnormalities." I think it would be medical professionals rather than scholars who would be aiming to prevent such abnormalities in patients with sepsis.

**Language Note:** The review process has identified that the English language must be improved. PeerJ can provide language editing services - please contact us at [email protected] for pricing (be sure to provide your manuscript number and title). Alternatively, you should make your own arrangements to improve the language quality and provide details in your response letter. – PeerJ Staff

Reviewer 1 ·

Basic reporting

The reviewer would like to congratulate the authors on an interesting study. The manuscript has mostly been well written, however there are some necessary changes for improving clarity of methods and flow of reading.

An overall comment is that the description of what was done and when is not very clear and needs to have more accurate language used throughout the manuscript for the reader to be able to better understand the method and results. This has been described in more detail below.

Comments on the Abstract:

Line 26: I suggest an expanded and more specific description of the study here. E.g. " This study explored whether LGG administration prior to and following induced sepsis could ameliorate cognitive deficits, and explored potential mechanisms.” I have suggested ameliorate instead of prevent, because a study designed to test only prevention would not give the probiotic after the surgery, and also a reduction in cognitive deficit was also of interest not just prevention. This suggestion has also been made for the same sentence in the introduction.

Line 31: Sentence beginning “the LGG-treated…”. This sentence needs to be reworded as it suggests that only the probiotic was continued after the surgery, and the vehicle given to the control group was not continued for another two weeks.

Line 32. For clarity, rather than just saying longitudinally, it would be better to more accurately describe when the testing was done.

Line 43: The final sentence is indicated causation but the study only shows an association. Reword sentence. E.g change “by preserving” to “and preserves.

Comments on the introduction:

Line 49: Sentence beginning “Sepsis is highly prevalent and affects..” Please clarify if this prevalence is per year or at one time? It should also be referenced.

Line 56: Sentence beginning "The pathogeneis". For better flow this sentence should be split into two sentences.

Line 60: Sentence beginning "Furthermore, the utilization of...". The word "in" should be deleted or alternatively it should say "causes changes in the gut microbiome"

Line 60: Sentence beginning "Furthermore, the utilization of...". This reference is not for an experimental study but is a review and does not prove cause. Wording should be less strong. I suggest writing "possibly contributing".

Line 62: Sentence beginning "A study suggested that". The referenced study shows causation and therefore the sentence would be clearer by deleting the first four words and beginning it with "treatment by fecal...."

Line 66. Change “attenuates” to “attenuated”, since reference is only showing results from one study so it is not yet known to always occur.

Line 69: Change “the common bacterium” to “a common bacterium” since it isn’t the only common one.

Line 75: I suggest an expanded and more specific description of the study here. E.g. " This study explored whether LGG administration prior to and following induced sepsis could ameliorate cognitive deficits, and explored potential mechanisms.” I have suggested ameliorate instead of prevent, because a study designed to test only prevention would not give the probiotic after the surgery, and also a reduction in cognitive deficit was also of interest not just prevention.

Comments on the method section:

Line 100: I suggest not using the word ‘modeling’. It would be clearer to be specific and refer to surgery or surgery & behaviour testing if that is what is being referred to.

Line 100. Please clarify on the timing of the surgery and behavioural tests for the different groups. The sentence beginning “the modelling was divided…” does not clearly explain the sequence. Did different mice and/or different groups have the behavioural tests done at different amounts of time after their surgery? E.g. some after 1 week and some after 3 weeks? Which mice had behavioural testing done? Please elaborate which mice were tested when and what groups they were in.
In this subsection it the timing of brain tissue sampling also needs to be explained. What timepoint and from which mice? Had the mice undergone behavioural testing? If so were they killed straight after or was there a gap?
All of this could possibly be outlined in a figure.

Line 111: Paragraph beginning “One week before modelling…”. It would be easier to follow the timeline of the LGG and vehicle administration if it is not split into so many sentences. Also the word injected is confusing a it is not typically used to describe gavaging (even though that is technically correct). And again, I suggest using term surgery rather than modelling.
I suggest something like “For one week prior to surgery, and two weeks following surgery, 200 μL bacterial solution (LGG) or and an equal volume of saline (vehicle) was administered to mice using a No. 12 gavage needle at the same time every day.”

Line 114: The timing of the behavioural tests needs to be more clear. Please elaborate on how many days post surgery the behavioural tests occurred, and also the timing of the tests relative to each other. Which was done first and were they done on the same day or separate days? How long was there between tests?

Line 127: For clarity I suggest changing “trained for the exposure” to “exposed to”.

Line 128. “subsequently”. Was this directly after the initial exposure to the objects or was it the following day?

Line 130: Were objects washed between each trial (between mice) or between each part of the trial (in between initial exposure and testing)?

Line 130: Sentence beginning “The mouse touched…’. Please reread and fix this sentence, it has words missing which make it unclear.

Line 162: It is unclear what the researchers are communicating with this statement. What was randomised and blinded and how?

Comments on results section:

It seems unnecessary to separate the effect of CLP from the effect of CLP + LGG in the results. It would be more logical to discuss the survival rate of all the groups in one subsection and the effects on behaviour/cognition in all the groups in another subsection.

Following on from this, Figure 1A is not behavioural testing and would therefore work better as a standalone figure, to go with the subsection on survival.

It seems unnecessary to have Figures 1B-1H as well as 2B-2H. Typically, these would be combined showing all the groups (as per figures 2B-2H) across time (as per figures 1B-1H). This would make it much easier to understand the results.

Figure 2A is out of place and would be better placed in methods.

In the description of behavioural results, it would be useful to indicate what the different results mean in the OFT, as has been done for NORT. Eg. Total distance indicates the activity level of the mice, and time in the centre/edge of box represents anxiety-like behaviour. Be careful to specify this rather than use the term cognitive deficits too much. Additionally, bear in mind that anxiety-like behaviour is not a cognitive deficit, it is an emotional behaviour.

Line 181: Just use term preference index since this has been explained in the methods.

Line196. Change “supplemented” to “supplementation”. Reword “reaching new project” as it is unclear what this means.

Line 197: reword “time spending on reaching new object and cognitive index” as it is unclear what this means.

Comment on discussion section:

Since the results show differences in cognition (recognition memory in NOR) but also anxiety-like behaviour (reduced time in centre in OFT), both of these terms/concepts should be used and discussed.

Line 236: Sentence beginning “A previous study has shown that probiotic..” is out of place and would best be moved to the first paragraph.

Line 274. Conclusion spelled incorrectly.

Line 276. Sentence beginning "LGG…" indicates causation. Despite this being a likely mechanism, the study only shows an association, it does not prove causation. Please reword.

Experimental design

The study seems to be well designed however because the description of what was done and when is unclear, it makes it difficult to assess this accurately. It will need to be reassessed once the method description has been improved.

Validity of the findings

No comment

Reviewer 2 ·

Basic reporting

Review for Wang et al

The study investigates the effects of Lactobacillus rhamnosus GG (LGG) on cognitive impairments in septic mice. It explores the impact of LGG on mice with sepsis-induced cognitive and emotional impairments, assessed using open field tests (OFT) and novel object recognition tests (NORT). Results indicate that septic mice treated with LGG showed improved cognitive functions and survival rates compared to untreated septic mice.

However, I have several major concerns with data analysis like counting viable neurons in the DG region of Nissl staining of Figure 3A, Figure 4 In the immunofluorescence data I do not see any difference in BDNF expression between CLP and CLP+LGG. Figure 5D is an inaccurate representation of data (quantification of western blot). This paper if published has several drawbacks in data analysis and misrepresentation of data.

Line 22. Italicize Lactobacillus rhamnosus in the abstract.
Line 69 Lactobacillus rhamnosus GG use an abbreviation instead of giving full form.
Line 80 Why did you take only female mice, isn’t an equal number of male and female mice should have been taken?
Figure 1A CLP+LGG causes more than 50% mortality in mice discuss the reason for this either in the main text or discussion.
Line 31 Cecal ligation and puncture (CLP) discusses how it can mimic sepsis for non-familiar readers in the introduction.
Line 208 You have abbreviated DG without giving its full form.
Line 212 Give the method of counting viable neurons in HE and Nissl staining in the method’s section did you use ImageJ software or the manual counting method?.
Line 214 What is CA1 and CA3? how the reader will know what it is if you do not give a brief description with reference.
Figure 3B Why the CLP region in DG staining looks like sham control.
Figure 3A and B The CA1 region in both HE and Nissl staining look the same, how you calculate a significant difference? in Fig 3D.
Figure 4 In the immunofluorescence data I do not see any difference between CLP and CLP+LGG. How do you conclude that BDNF expression was significantly decreased in CA1 and DG regions of CLP.

Figure 5 Legends There is no (D)

Figure 6 Your quantification of the BDNF and p-TrkB/TrkB looks wrong. I doubt this data and quantification.

Experimental design

Figure 4 In the immunofluorescence data I do not see any difference between CLP and CLP+LGG. How do you conclude that BDNF expression was significantly decreased in CA1 and DG regions of CLP.


Figure 6 Your quantification of the BDNF and p-TrkB/TrkB looks wrong. I doubt this data and quantification.

Validity of the findings

Figure 3B Why the CLP region in DG staining looks like sham control.
Figure 3A and B The CA1 region in both HE and Nissl staining look the same, how you calculate a significant difference? in Fig 3D.
Figure 4 In the immunofluorescence data I do not see any difference between CLP and CLP+LGG. How do you conclude that BDNF expression was significantly decreased in CA1 and DG regions of CLP.


Figure 6 Your quantification of the BDNF and p-TrkB/TrkB looks wrong. I doubt this data and quantification.

---

## Round 0.2 · Minor Revisions

As both reviewers have noted, the authors have addressed the previous comments well and as a result the manuscript is markedly improved. If they can attend to the few remaining minor comments from Reviewer 1, I believe the manuscript will be acceptable for publication.

Reviewer 1 ·

Basic reporting

The manuscript and figures are much improved from the first draft, and now clearly explains the study methodology and results. I have only minor suggestions:

Line 45. This sentence still didn’t have a timeframe for the numbers stated and therefore didn’t actually make sense. I looked up the reference and suggest the sentence: “Sepsis is highly prevalent with a high mortality rate, e.g. in 2017 it affected 48.9 million people worldwide and represented almost 20% of global deaths (Rudd et al., 2020).
Line 48: Sepsis survivors may or may not experience neurocognitive dysfunction, and it may or may not be permanent. Perhaps reword to “Sepsis survivors can experience short or long term neurocognitive dysfunction”.
Line 63: Sentence “However, the current research on sepsis and SAE mainly focuses on male mice. Few studies on sepsis and SAE have been performed in female mice, which is an important gap in the current research as gender differences may affect learning and memory behavior (Fleischer & Frick, 2023)” doesn’t fit it here very well. With the “however” it seems like it is a rebuttal to the previous statement, and the sudden change of topic from mechanism to research design doesn’t flow. I suggest changing the word however to additionally, and moving the sentence to Line 72, before the sentence beginning “this study explored…”
Line 79: Please add in the total number of mice.
Line 181: All text in the sentences “The mice in all three….is plotted in Figure 1A” don’t fit in this section and should be moved to the methods section, probably at the end of the treatments subsection.
Line 199 – Line 203: Sentence beginning “this reflects..” I feel this sentence is the only summary and doesn’t say much. You can be much conclusion about the findings. There was increased time and distance in the inner area of the OFT, in the CLP mice, which indicates the CLP_sepsis caused increased emotionality, and this was reduced/ameliorated in the mice with LGG supplementation. This is an interesting finding and should be clearly stated. It could be included here, or added to line 213 with the other conclusion sentence.
Line 271: typically the term “microbiome-gut-brain axis” is used.

Experimental design

No additional comments

Validity of the findings

No additional comments

Reviewer 2 ·

Basic reporting

I appreciate the authors for their prompt and thorough revisions addressing my earlier concerns. The modifications have significantly strengthened the manuscript, and I am now satisfied with the completeness and clarity of the content.

Experimental design

no comment

Validity of the findings

no comment

---

## Round 0.3 · accepted · Accept

The authors have now addressed all of the review comments and I believe the manuscript is now acceptable for publication.